# Donor Screening Revisions of Fecal Microbiota Transplantation in Patients with Ulcerative Colitis

**DOI:** 10.3390/jcm11041055

**Published:** 2022-02-17

**Authors:** Xiaochen Zhang, Dai Ishikawa, Kei Nomura, Naoyuki Fukuda, Mayuko Haraikawa, Keiichi Haga, Tomoyoshi Shibuya, Toshihiro Mita, Akihito Nagahara

**Affiliations:** 1Department of Gastroenterology, Juntendo University School of Medicine, 2-1-1 Hongo, Bunkyo-ku, Tokyo 113-8421, Japan; x.zhang.as@juntendo.ac.jp (X.Z.); ke-nomura@juntendo.ac.jp (K.N.); m.haraikawa.bk@juntendo.ac.jp (M.H.); khaga@juntendo.ac.jp (K.H.); tomoyosi@juntendo.ac.jp (T.S.); nagahara@juntendo.ac.jp (A.N.); 2Department of Intestinal Microbiota Therapy, Juntendo University School of Medicine, 2-1-1 Hongo, Bunkyo-ku, Tokyo 113-8421, Japan; 3Department of Tropical Medicine and Parasitology, Faculty of Medicine, Juntendo University, 2-1-1 Hongo, Bunkyo-ku, Tokyo 113-8421, Japan; n-fukuda@juntendo.ac.jp (N.F.); tmita@juntendo.ac.jp (T.M.)

**Keywords:** fecal microbiota transplantation, donor screening, ulcerative colitis

## Abstract

Fecal microbiota transplantation (FMT) has been recognized as a promising treatment for dysbiosis-related diseases. Since 2014, FMT has been utilized to treat ulcerative colitis (UC) in our clinical studies and has shown efficacy and safety. As donor screening (DS) is the primary step to ensure the safety of FMT, we report our experience with DS and present the screening results to improve the prospective DS criteria and provide references for future studies. The donor candidates were screened according to the DS criteria. The first DS criteria were proposed in June 2014 and revised substantially in May 2018. We further sorted the screening results and costs of laboratory tests. From June 2014 to April 2018, the DS eligibility rate was 50%. The total laboratory testing cost for each candidate was JPY 17,580/USD 160.21. From May 2018 to September 2021, the DS eligibility rate was 25.6%. The total laboratory testing cost for each candidate was JPY 40,740/USD 371.36. The reduction in donor eligibility rates due to more stringent criteria should be considered for cost and safety. Studies must consider the latest updates and make timely modifications in the DS criteria to ensure patient safety.

## 1. Introduction

Fecal microbiota transplantation (FMT), a therapeutic regimen for transplanting healthy donor feces to reprogram the recipient’s gut microbiota, has been proposed as a promising treatment for a wide range of dysbiosis-related diseases [1,2,3,4,5]. Moreover, FMT has a well-established role in the treatment of recurrent *Clostridioides*
*difficile* infection (CDI) in the West [6,7,8], with an overall cure rate of approximately 90% [2,9,10].

Ulcerative colitis (UC) is a refractory chronic inflammatory bowel disease (IBD), and standard drugs are not always effective for its treatment [11]. Although the pathogenesis of UC is not yet fully understood, previous reports have shown that dysbiosis caused by a decline in the diversity and abundance of intestinal microbiota in patients with UC is related to the development of the aberrant immunological response observed in IBD cases [12,13,14]. In a meta-analysis of four randomized controlled trials, the clinical remission rate of patients with UC who had undergone FMT was 42.1%, which was significantly higher than that of the control group, which was 22.6% [15]. Therefore, because of its efficacy, FMT has been studied extensively as a promising treatment for UC.

With the continuous in-depth research and establishment of FMT as a reliable therapeutic outlet, a great demand for fecal donors has been generated. However, it is well-known that pathogenic organisms can be transmitted through fecal transplantation. In fact, it has been reported that FMT may transmit cytomegalovirus (CMV) through urine-stained feces from an unscreened donor [16]. Moreover, in November 2019, the United States reported two cases of insufficient screening of *Escherichia coli* where one of the patients died because of bacteremia caused by drug-resistant *E. coli* transmission [17]. Therefore, a safe source of feces as a transplant material is the primary checkpoint for FMT. However, the screening criteria for potential infectious pathogens have always been inconsistent. Institutions in different countries have proposed their donor screening (DS) criteria for FMT. OpenBiome, the longest-standing international stool bank, has released the latest DS criteria in 2021 [18], but the European Consensus Conference [19], as well as other countries including Australia and Canada, have also proposed their own DS criteria [20,21,22]. This is because the required standards of DS vary between national health regulatory agencies. Therefore, the DS criteria should be adjusted according to the national socio-cultural context. The Chinese FMT working group reported that after two revisions to strengthen DS, the incidence of adverse events dropped from 30.7% to 20.1% [23]; thus, strict DS security control could prevent harmful consequences.

Since 2014, we have been studying the implementation of FMT in the treatment of UC and have reported the efficacy and safety [24,25,26]. Furthermore, we have constantly improved our DS criteria to ensure patient safety. Our first DS criteria (2014 version), which were formulated according to the Amsterdam protocol, were employed from June 2014 to April 2018 with some minor modifications [27,28,29]. Based on these criteria, we completed three clinical studies [24,25,26]. From May 2018, tremendous changes have been made to enhance safety, and these revised criteria (2018 version) are being utilized to perform a clinical study. Therefore, we report our experience and propose the DS criteria used in our hospital for improving the prospective DS criteria and providing a reference for future studies.

## 2. Materials and Methods

The DS process (Figure 1) consisted of three steps: (1) pre-screening, (2) formal screening, and (3) final screening. Pre-screening included donor candidate recruitment, preliminary assessment, and a questionnaire survey. Formal screening refers to laboratory testing. Final screening included reassessment of health conditions and enrollment of the eligible donor.

### 2.1. Pre-Screening

Those donor candidates, recommended to the study by patients or college student volunteers from the Juntendo University School of Medicine, took the initiative to participate after learning about the study and were then recruited. The recruitment of candidates was not restricted by sex. The preliminary candidate assessment was conducted without fixed criteria at the Juntendo University Hospital from June 2014 to September 2021, and it was mainly comprised of inquiries regarding general health conditions, such as any discomfort or illnesses the candidates recently had. Candidates who were deemed to have no abnormalities by a physician were then eligible to participate in the questionnaire survey.

Concerning the demographic information in the questionnaire survey, the age [30] and body mass index (BMI) of the donor candidates are now required by the 2018 revised questionnaire. In addition, compared with the medical history checked in the 2014 version, the 2018 version further clarified which intestinal microbiota-related diseases should be excluded [31,32]. Furthermore, a family history of colorectal carcinoma or active gastrointestinal infection has to be disclosed by the donor candidates. Additional non-infectious-related clinical information, including the use of antimicrobials or protein pump inhibitors (PPI), and the presence of psychiatric symptoms [33,34,35] are now required under the 2018 revision. Finally, the risk investigation of infectious diseases is also more detailed than in the 2014 version. The specific list of questionnaires employed as donor exclusion criteria can be found in Table 1. Candidates who met any of these criteria were excluded.

### 2.2. Formal Screening

To minimize the risk of infection transmission, only candidates who passed the pre-screening step underwent laboratory tests, including fecal and blood testing. The feces and blood samples of the donor candidates were sent to the laboratory of the SRL Company (Tokyo, Japan) that specializes in clinical laboratory testing, and comprehensive testing was performed in accordance with our fecal and blood screening criteria.

The fecal test in the 2014 version (Table 2) included an examination of parasitic ova and cysts, while in the test of the 2018 revision (Table 2) the parasite bodies, *cryptosporidium*, *giardia*, and *Entamoeba histolytica* were added. In particular, *Entamoeba histolytica*, combined with a clinical evaluation, was used for the diagnosis of *Entamoeba histolytica* in the 2018 version. As the use of A and B toxins of *C. difficile* alone may be unreliable for CDI diagnosis in a proportion of patients [36], dual testing of the toxin and the glutamate dehydrogenase (GDH) protein is now employed to improve diagnostic accuracy. In addition, *E. coli* verotoxin testing has been performed since June 2019, and further screening of all *E. coli* strains has been performed promptly from August 2020 because of safety alerts (12 March 2020) issued by The U.S. Food and Drug Administration (FDA) [37]. Screening for norovirus and rotavirus, which are transmitted through the fecal-oral route and can easily cause diarrhea, as well as fecal occult blood testing, which is often used to evaluate intestinal health and to screen for colorectal cancer, have been included [38].

The blood test items of the 2014 version (Table 3) mainly included basic infectious diseases. In 2018 (Table 3), testing for the hepatitis E virus and immunoglobulin M antibodies against the Epstein-Barr virus has been included. Initially, parasite positivity was solely determined through microscopic examination of stools. However, this method depends on the inspector’s subjective judgment, which can potentially result in a misdiagnosis. Therefore, a parasite-specific serum antibody screening kit [39] is now additionally employed to determine positivity to the following parasites: *Dirofilaria immitis*; *Ascaris suum*; *Anisakis*; *Gnathostoma spinigerum*; *Cysticercus cellulosae*; *Strongyloides stercoralis*; *Paragonimus westermanii*; *Clonorchis sinensis*; *Sparganum mansoni*; *Toxocara canis*; *Paragonimus miyazakki*; and *Fasciola hepatica*. Furthermore, polymerase chain reaction (PCR) testing was performed [40,41] for the reconfirmation of positive *S. stercoralis* results. The risk of CMV reactivation in patients with severe UC ranges from 21% to 34% [42,43]. Since May 2018, the parameters for testing of CMV have been revised twice, continuously strengthening the specificity and accuracy of virus detection to reduce the possibility of re-exacerbation. Further, severe acute respiratory syndrome coronavirus 2 (SARS-CoV-2) testing has been implemented from July 2020. Finally, complete blood count tests, as well as liver and kidney function examinations were conducted to assess health conditions. The complete list of clinical infectious information employed as donor exclusion criteria can be found in Table 2 and Table 3. Candidates meeting any of these criteria were excluded.

### 2.3. Final Screening

The absence of any health conditions, such as diarrhea and fever, was ascertained during the days prior to stool submission. The health conditions of the cohabiting family members were also assessed [44]. The candidates could voluntarily donate feces multiple times for three months, and the donations were processed and cryopreserved at –80 °C for up to six months [45].

## 3. Results

### 3.1. Eligible Donor Screening from 2014 to 2018

A total of 138 candidates came to our hospital from June 2014 to April 2018 for preliminary assessment, and 61 were excluded at this stage. Most of the remaining 77 candidates were blood relatives (*n* = 44) or spouses (*n* = 26) recommended by patients, and a small number of candidates were volunteers (*n* = 7). The ratio of volunteers to candidates who passed the preliminary assessment was 9.1% (7/77). This cohort comprised 30 men and 47 women, with an average age of 42.2 years (range, 20–68 years). After questionnaire completion, three candidates were excluded because of their medical history, while one patient was also excluded of acute diarrhea that had been present for the previous three months.

Laboratory testing was performed on 73 candidates. Of these, one individual was ruled out due to *C. difficile* toxin-positive feces, and three candidates were rejected because of CMV-positive blood testing. The remaining 69 candidates and their cohabitants were all reconfirmed to be in healthy condition. Therefore, 50.0% of the initial volunteers passed the DS process. The final cohort included 27 men and 42 women, with an average age of 41.9 years (range, 20–68 years) (Figure 2).

The total cost of laboratory testing for performing fecal (JPY 3650/USD 33.27) and blood tests (JPY 13,930/USD 126.94) was JPY 17,580/USD 160.21.

### 3.2. Eligible Donor Screening from 2018 to 2021

Among the 117 candidates who visited our hospital between May 2018 and September 2021, 43 were excluded after preliminary assessment. The remaining 74 candidates comprised volunteers (*n* = 43) and patients’ blood relatives (*n* = 28) and spouses (*n* = 3). The ratio of volunteers to candidates who passed the preliminary assessment was 58.1% (43/74). This cohort included 43 men and 31 women, with an average age of 33.2 years (range, 19–70 years). After questionnaire completion, one, two, one, one, three, and two candidates were excluded because of a family history of colorectal carcinoma, BMI > 25 kg/m^2^, received antimicrobial therapy within the previous three months, received probiotics or PPI therapy within the previous three months, vaccinations received in the previous three months, and high-risk sex performed in the previous three months, respectively.

Laboratory testing was performed on the remaining 64 candidates. A total of eight candidates showed abnormal fecal test results, twenty-one had abnormal blood test results, and another eight showed abnormalities in both blood and fecal tests. The details of the abnormal testing results are described in Appendix A.

A total of 20 (31.3%) candidates were excluded due to positive results on parasite antibody testing, more prominent than other laboratory testing items (Table 4). The most common parasite detected was *S. stercoralis* (55.0%, *n* = 11; Table 5), but PCR testing confirmed that all the participants were negative. Of these, the stools of three candidates (Patients 2, 5, and 7) who showed negative results with laboratory tests, except for *S. stercoralis*, were nonetheless utilized for FMT (Appendix A).

After reconfirming the health status of the candidates and their cohabitants, 30 candidates were included in the final cohort. Therefore, 25.6% of the initial candidates passed the screening, including 18 men and 12 women, with an average age of 36.0 years (range, 20–69 years) (Figure 3).

The total cost of laboratory testing for performing fecal (JPY 14,260/USD 129.99) and blood testing (JPY 26,480/USD 241.37) was JPY 40,740/USD 371.36.

## 4. Discussion

Since 2014, no serious adverse events have been noted in patients admitted to our studies, regardless of the use of either the 2014 or the 2018 version of the DS criteria. However, other institutions have reported serious adverse events due to insufficient DS [16,17]. Therefore, it is necessary to continuously strengthen the screening of infectious diseases and minimize the occurrence of adverse events. To the best of our knowledge, this is the first report in Japan that describes the screening items, costs, and results to provide reference for future studies.

It has been shown that the microbial diversity of the donor is the main factor affecting the efficacy of FMT for patients with UC [46,47]. In addition, feces donated by siblings provide a greater long-term remission rate than those obtained from a parent or an offspring [26]. However, the current evidence concerning the features of optimal donors for UC treatment remains limited. In addition, in our study, although patients were more receptive to transplanting feces from donors recommended by the patients themselves, this caused some inconveniences. For example, as the screening process is quite time-consuming, the patient’s therapeutic opportunity may be delayed; there is also a risk of revealing the donor’s personal medical history. A previous report showed that there is no significant difference in the remission rate of CDI between patients who have undergone FMT with feces from a patient-recommended donor and those who have been transplanted with undirected donor materials [48]. Therefore, many institutions have adopted the method of recruiting volunteers as donor candidates [18,20,49,50]. We have also strengthened the recruitment of undirected donor volunteers after revising the DS criteria in 2018. The ratio of volunteers to candidates who passed the preliminary assessment increased from 9.1% (7/77) to 58.1% (43/74). In this study, most of the recruited volunteers were medical students who had medical knowledge and, thus, could better understand our recruitment requirements. Overall, although the questionnaire has become more rigorous from May 2018, the pre-screening pass rates did not decrease (Figure 2 and Figure 3). The main reason for this may be related to the change in the target population from which the recruited candidates are selected.

In 2015, the infection rate following parasite testing in Tokyo using the feces of school students as samples was <10% and showed a downward trend [51]. However, the parasite-specific antibody screening method employed in this study produced unexpectedly high positive results (31.3%). All the positive patients had the lowest evaluation of +1 on a 4-point positive scale, which is normally considered a false positive. We hypothesized that non-specific reactions would be the cause of this, although it has not been confirmed because only testing for *S. stercoralis* was conducted. In fact, the PCR test revealed that all 11 candidates were negative for *S. stercoralis*. Three candidates who had originally tested positive for *S. stercoralis* were nonetheless selected based on their epidemiological information and background [52]. Although several PCR inaccuracies for the diagnosis of *S. stercoralis* have been reported, PCR may have a confirmatory test effect [53]. In our study, the unnecessary exclusion due to *S. stercoralis* was also reduced by PCR. In addition, some of the parasites included in the screening method, such as *Gnathostoma spinigerum* and *Anisakis*, are unlikely to infect the recipient through FMT. Taken together, the current screening methods of parasites may cause the unnecessary exclusion of some donor candidates, and detection technology for parasites still needs to be further optimized.

There were limitations to this study. First, we did not record the reasons why candidates were excluded at the preliminary assessment stage. Second, this was a single-center clinical study, and volunteers were recruited non-randomly. Third, the current parasite screening guidelines may cause unnecessary exclusion of candidates; thus, pass rates may not be representative.

In the ongoing clinical study, we have begun to evaluate the enteric pathogens in the fecal solution immediately after the donor submits the feces by using the FilmArray gastrointestinal panel (BioFire Diagnostics, Salt Lake City, UT, USA), which distinguishes possible causes of gastroenteritis (seven bacteria, four parasites, five viruses, and three diarrheagenic *E. coli* and *Shigella*) to strengthen the response to the safety alert from the FDA (12 March 2020). We also aim to use a highly sensitive intestinal inflammatory marker, calprotectin, in the next study [54]. We hope to always propose the most optimized DS criteria to support future FMT studies in different emerging fields.

## 5. Conclusions

Screening stool donors for successful FMT in patients with UC is challenging. Researchers in this field should always consider the latest microbiota studies and global epidemic status to allow for timely revisions of the DS criteria to minimize the occurrence of adverse events. In addition, the reduction in the donor eligibility rates due to more stringent criteria should be considered in light of the balance between cost and safety.

## Figures and Tables

**Figure 1 jcm-11-01055-f001:**
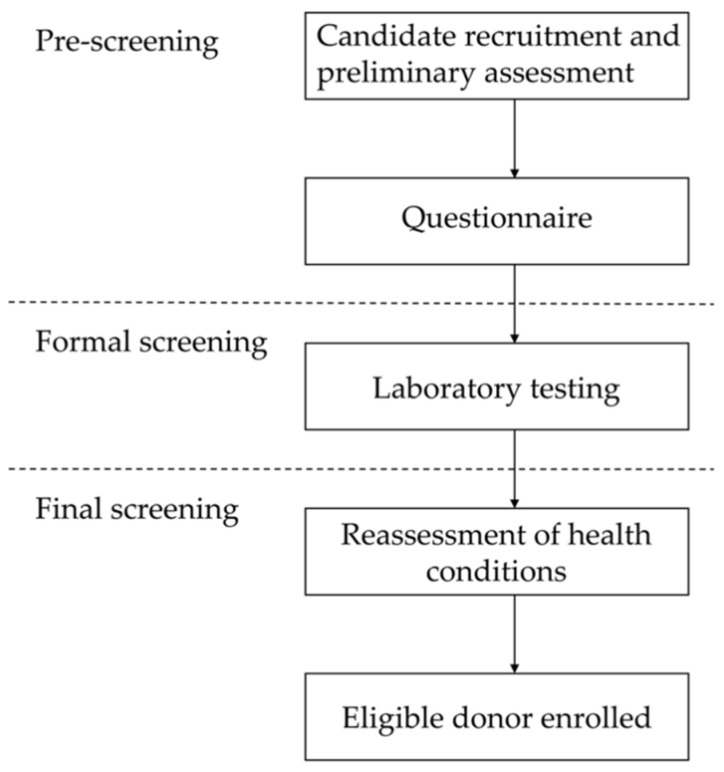
Flowchart representing the screening process for recruiting eligible candidates.

**Figure 2 jcm-11-01055-f002:**
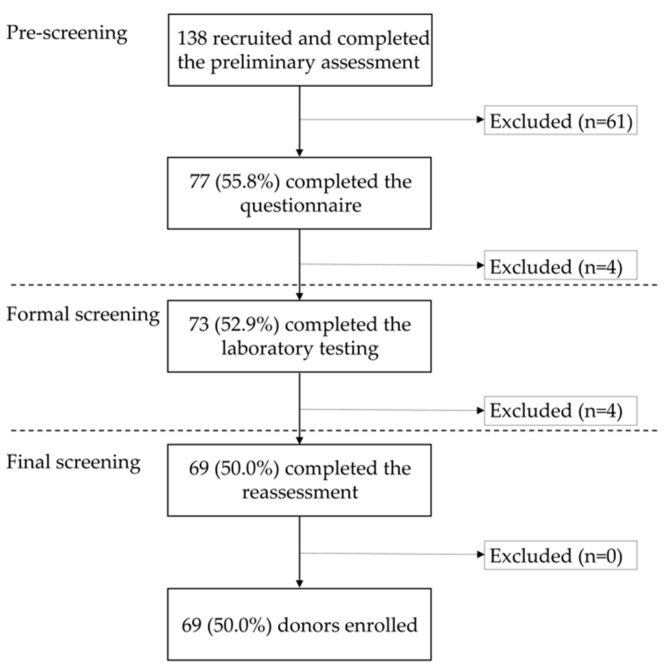
The process and results of donor screening from June 2014 to April 2018.

**Figure 3 jcm-11-01055-f003:**
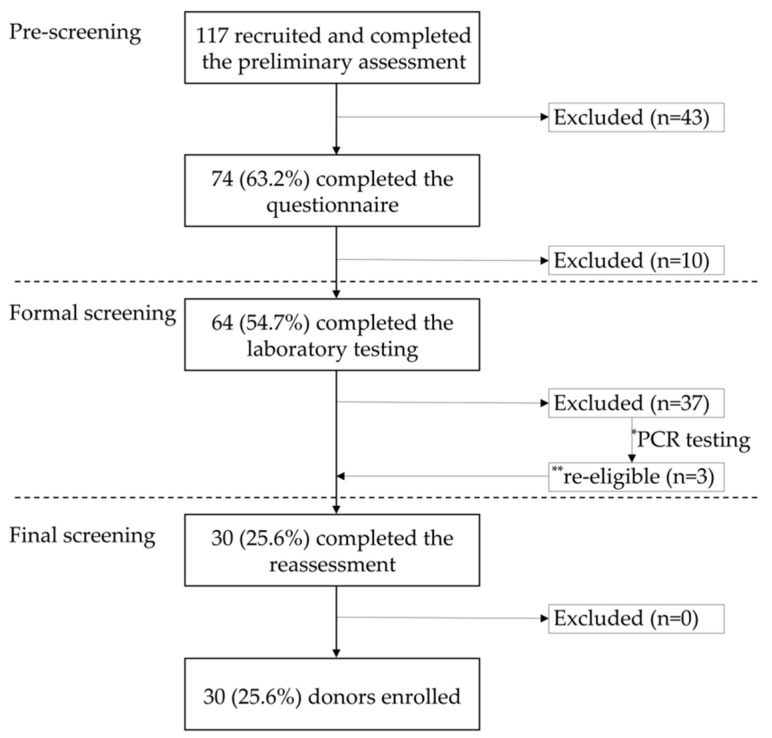
The process and results of donor screening from May 2018 to September 2021. * 11 candidates with positive antibody test results for *Strongyloides stercoralis* were tested again by PCR; ** three candidates with negative PCR results for *S. stercoralis* infection were declared eligible. PCR: polymerase chain reaction.

**Table 1 jcm-11-01055-t001:** Donor screening criteria: questionnaire.

Questionnaire (6/2014–4/2018)	Questionnaire (5/2018–9/2021)
Factor affecting the composition of the intestinal microbiota:	Factor affecting the composition of the intestinal microbiota:
Past medical history: severe diseases, malignancies, surgery, and hospitalization	Irritable bowel syndrome, chronic constipation, and chronic diarrhea
	Intrinsic gastrointestinal illness: inflammatory bowel disease and colonic polyps
	Autoimmune disease
	Atopic disease (including atopic dermatitis)
	Chronic fatigue syndrome
	History of major gastrointestinal surgery or malignant disease
	Any psychiatric disorder assessed by the HAMD ^2^ or NPI ^3^ (8/2020–present)
Probiotic consumption in the last 3 months	Consumption of antimicrobials (antibiotics, antivirals, and antifungals), probiotics, or PPIs ^4^ in the last 3 months
Taking any medications	Taking any medications, any active medical illness, or symptoms
Acute diarrhea in the last 3 months	Acute diarrhea in the last 3 months
	Family member with colorectal carcinoma or active gastrointestinal infection
	BMI ^1^ < 18 or >25 kg/m^2^ or metabolic syndrome
	Age < 18 or >70 years
Risk investigation of infectious disease:	Risk investigation of infectious disease:
International travel to an area with high risk of traveler’s diarrhea in the last 6 months	International travel to an area with high risk of traveler’s diarrhea in the last 6 months
High-risk sex in the last 3 months	High-risk sex in the last 3 months
	Tattoo, body piercing, or acupuncture in the last 6 months
	Needlestick accident in the last 6 months
	Household members with active gastrointestinal infection
	History of vaccination with a live attenuated virus in the last 3 months
	Incarceration or a history of incarceration
	Known history of infectious diseases (i.e., HIV ^5^, hepatitis, syphilis, tuberculosis, among others)

^1^ BMI: body mass index; ^2^ HAMD: Hamilton Depression Scale; ^3^ NPI: Neuropsychiatric Inventory; ^4^ PPIs: proton pump inhibitors; ^5^ HIV: human immunodeficiency virus.

**Table 2 jcm-11-01055-t002:** Donor screening criteria: fecal test.

Fecal Test (6/2014–4/2018) Test Item	Cost per Person	Fecal Test (5/2018–9/2021) Test Item	Cost per Person
Ova and cysts of parasites	¥400/$3.65	Parasites, ova, cysts, *Cryptosporidium*, *Giardia*, and *Entamoeba histolytica*	¥1300/$11.85
*Clostridioides**difficile* toxin	¥1250/$11.39	*Clostridioides**difficile* toxin/*Clostridioides d**ifficile*-specific GDH ^1^	¥1250/$11.39
Microscopy and culture:	¥2000/$18.23	Microscopy and culture:	¥2300/$20.97
*Salmonella*		*Salmonella*	
*Shigella*		*Shigella*	
*Yersinia*		*Yersinia*	
*Campylobacter*		*Campylobacter*	
		*Escherichia coli* (8/2020–present)	
		Diarrheagenic *Escherichia coli*	
Enterohemorrhagic *Escherichia coli*		Enterohemorrhagic *Escherichia coli*	
Other common enteric pathogens: Vancomycin-resistant Enterococcus Methicilin-resistant *Staphylococcus aureus* Carbapenem-resistant Enterobacteriaceae Multidrug-resistant Gram-negative bacteria, etc.		Other common enteric pathogens: Vancomycin-resistant Enterococcus Methicilin-resistant *Staphylococcus aureus* Carbapenem-resistant Enterobacteriaceae Multidrug-resistant Gram-negative bacteria, etc.	
		*Escherichia coli* verotoxin (6/2019–present)	¥4000/$36.46
		Norovirus	¥4000/$36.46
		Rotavirus	¥1210/$11.03
		Fecal occult blood testing	¥200/$1.82
Total cost per candidate	¥3650/$33.27	Total cost per candidate	¥14,260/$129.99

^1^ GDH: glutamate dehydrogenase. ¥1 JPY = $0.009 USD.

**Table 3 jcm-11-01055-t003:** Donor screening criteria: blood test.

Blood Test (6/2014–4/2018)		Blood Test (5/2018–9/2021)	
Test Item	Cost per Person	Test Item	Cost per Person
Infection:		Infection:	
Hepatitis A virus antibody, Ig ^1^ M	¥800/$7.29	Hepatitis A virus antibody, IgM	¥800/$7.29
Hepatitis B virus surface antigen/antibody Hepatitis B virus core antibody	¥1100/$10.02 ¥830/$7.56	Hepatitis B virus surface antigen/antibody Hepatitis B virus core antibody	¥1100/$10.02¥850/$7.74
Hepatitis C virus antibody	¥1000/$9.11	Hepatitis C virus antibody	¥1000/$9.11
		Hepatitis E virus antibody, IgA	¥1320/$12.03
HIV ^2^ antibody and antigen	¥800/$7.29	HIV antibody and antigen	¥800/$7.29
Human T-cell lymphotropic virus-1 antibody	¥750/$6.84	Human T-cell lymphotropic virus-1 antibody	¥750/$6.84
Syphilis (RPR/TP)	¥850/$7.75	Syphilis (RPR ^4^/TP ^5^)	¥850/$7.75
CMV ^12^	¥500/$4.56	CMV antigen pp65 (5/2018–5/2021)/RT-PCR ^6^ (6/2021)	¥2000/4090/$18.23/37.28
Tuberculosis (IFN ^3^-γ)	¥5300/$48.30	Tuberculosis (IFN-γ)	¥4500/$41.0
*Entamoeba histolytica* antibody	¥2000/$18.23		
		Epstein-Barr virus IgM	¥1000/$9.11
		Parasite-specific antibody screening test	¥2500/$22.79
		SARS-CoV-2 ^7^ Antibody (7/2020–5/2021)/antigen (6/2021–present)	¥900/6000/$8.20/54.69
		Health condition:	
		Complete blood count	¥200/$1.82
		Electrolytes (sodium, potassium, and chlorine)	¥240/$2.19
		Renal function tests (blood urea nitrogen and creatinine)	¥180/$1.64
		Liver function tests (AST ^8^, ALT ^9^, ALP ^10^, and γ-GT ^11^)	¥200/$1.82
		Albumin	¥50/$0.46
		C-reactive protein	¥50/$0.46
Total cost per candidate	¥13,930/$126.94	Total cost per candidate	¥26,480/$241.37

^1^ Ig: immunoglobulin; ^2^ HIV: human immunodeficiency virus; ^3^ IFN: interferon; ^4^ RPR: rapid plasma regain; ^5^ TP: *Treponema pallidum*; ^6^ RT-PCR: reverse transcription-polymerase chain reaction; ^7^ SARS-CoV-2: severe acute respiratory syndrome coronavirus 2; ^8^ AST: aspartate transaminase; ^9^ ALT: alanine transaminase; ^10^ ALP: alkaline phosphatase; ^11^ γ-GT: gamma-glutamyl transferase; ^12^ CMV: Cytomegalovirus. ¥1 JPY = $0.009 USD.

**Table 4 jcm-11-01055-t004:** Positive rate of laboratory testing items (5/2018–9/2021).

	Fecal Testing (N = 64)				Blood Testing (N = 64)				
Test Item	Fecal Occult Blood Testing	*Clostridioides difficile*-Specific GDH ^2^	*Escherichia Coli*	Other Enteric Pathogens	Parasites	Renal Function	Liver Function	Epstein–Barr Virus Ig ^3^ M	CRP ^4^
N ^1^	2	1	11	3	20	2	7	2	2
Positive rate	3.1%	1.6%	17.2%	4.7%	31.3%	3.1%	10.9%	3.1%	3.1%

^1^ N indicates the number of candidates who showed an abnormal test item. ^2^ GDH, glutamate dehydrogenase; ^3^ Ig, immunoglobulin; ^4^ CRP, C-reactive protein.

**Table 5 jcm-11-01055-t005:** Positive rate of parasite-specific antibody screening testing items.

	*Dirofilaria immitis*	*Ascaris suum*	*Anisakis*	*Gnathostoma spinigerum*	*Cysticercus cellulosae*	*Strongyloides stercoralis*	*Paragonimus westermanii*	*Clonorchis sinensis*	*Sparganum mansoni*
N ^1^	1	2	2	7	1	11	1	4	1
Positive rate	5%	10%	10%	35%	5%	55%	5%	20%	5%

^1^ N indicates the number of candidates who showed an abnormal parasite-specific antibody among the screening items. Positive rate indicates proportion of N out of 20 candidates with positive parasite-specific antibody screening results.

## Data Availability

Not applicable.

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
