# Peer review of "Donor Screening Revisions of Fecal Microbiota Transplantation in Patients with Ulcerative Colitis"

_jcm, 2022, doi:10.3390/jcm11041055_

Round 1

Reviewer 1 Report

Zhang et al. report here their experience of donor screening and selection in clinical trials assessing Fecal Microbiota Transplantation (FMT) to treat ulcerative colitis (UC). Interestingly, by comparing donor screening criteria and modalities between two periods of time (2014-18 vs 2018-21), the authors show how FMT research evolved these last decade with increasing parameters now assessed for donor selection, increasing involvement of healthy volunteers in place of patient's relatives and screening issues regarding to unspecific laboratory test especially with parasitical serologies. Costs descriptions constitute also an interesting aspect of this study despite, these are difficult to transpose from one country to another as discussed by the authors.

Major limitations:

  • The authors suggest that the donor screening they proposed may constitute a standard for similar FMT studies. However, some important laboratory tests are missing and may lower the external validity of this protocol. For example, fecal test for multi-drug resistant bacteria involving not only drug-resistant E. coli (e.g. vancomycin - resistant enterococcus or methicilin-resistant staphylococcus aureus, etc) and fecal calprotectin measurement should be implemented.
  • The authors report that CMV transmission through FMT may be possible and insist on that point. However fecal transmission of CMV is still a matter of debate and CMV screening involving CMV blood test as proposed by the authors are not proven to be related to pathogenic CMV fecal excretion. This point should be discussed.
  • English language should be significantly improved as some sentences may be confusing.

Minor limitations

  • Clostridium difficile should be replaced by Clostridioides difficile
  • Table 4 and 5 are very difficult to read and should be simplified
  • "Other common enteric pathogens" screened in feces (Table 2) should be specified
  • Many redundancies are present in the discussion and especially, the paragraph about the cost of donor screening may benefit from a rewriting.
  • Authors mentioned "the expected efficacy" of FMT in UC, however exact FMT procedure and efficacy are still matter of debate and intensive research. This sentence may be nuanced.

Author Response

Question: The authors suggest that the donor screening they proposed may constitute a standard for similar FMT studies. However, some important laboratory tests are missing and may lower the external validity of this protocol. For example, fecal test for multi-drug resistant bacteria involving not only drug-resistant E. coli (e.g. vancomycin - resistant enterococcus or methicilin-resistant staphylococcus aureus, etc) and fecal calprotectin measurement should be implemented.

Response: The authors would like to thank the reviewer for his/her constructive critique to improve the manuscript. We have made every effort to address the issues raised and to respond to all comments. The revisions are indicated in red font in the revised manuscript. Please, find next a detailed, point-by-point response to the reviewer's comments. We hope that our revisions would meet the reviewer’s expectations.

Since August 2020 we have checked all E. coli strains (presented in Table 2b) and excluded donor candidates who were E. coli-positive, including those with drug-resistant E. coli. Please note that we have revised the corresponding part in the main text to present this information clearer. The revised part is as follows:

“In addition, E. coli verotoxin testing has been performed since June 2019, and further screening of all E. coli strains has been performed promptly from August 2020 because of safety alerts (March 12, 2020) issued by The U.S. Food and Drug Administration (FDA).” (Lines 114–117)

Moreover, we have included vancomycin - resistant enterococcus, methicilin-resistant Staphylococcus aureus, carbapenem-resistant Enterobacteriaceae, and Multidrug-resistant Gram-negative bacteria, as “Other common enteric pathogens” in Table 2. We have also filled the corresponding content in Table 2.

Finally, we did not perform the calprotectin test to the donor candidates, mainly because we considered that the patient's disease history had been asked in the questionnaire and the possibility of intestinal inflammation was excluded. Indeed, fecal occult blood testing and complete blood count can also assist in the diagnosis of intestinal inflammation. The reviewer’s suggestion was of great importance to strengthen the safety of our screening criteria. Thus, we decided to include the calprotectin test in our next study. We have discussed this issue in the revised manuscript as follows:

“We also aim to use a highly sensitive intestinal inflammatory marker, calprotectin, in the next study [54].” (Lines 267–268)

Question: The authors report that CMV transmission through FMT may be possible and insist on that point. However fecal transmission of CMV is still a matter of debate and CMV screening involving CMV blood test as proposed by the authors are not proven to be related to pathogenic CMV fecal excretion. This point should be discussed.

Response: We apologize to the reviewer for the confusion.

We aimed to state that since CMV is shed at high level in the urine, this patient may have had CMV infection by transplanting the urine-stained feces of his young child. To avoid confusion, we have revised the corresponding part in the Introduction section as follows:

“In fact, it has been reported that FMT may transmit cytomegalovirus (CMV) through urine-stained feces from an unscreened donor [16].” (Lines 46–48)

Besides, patients with UC are more prone to CMV reactivation or infection due to impaired intestinal immune barrier and exposure to multiple immunosuppressive agents. Therefore, to protect the safety of patients and prevent the urine-stained feces from being transplanted into the intestine of patient, causing re-exacerbation, we conducted a CMV test. We have presented this information in the revised manuscript as follows:

“The risk of CMV reactivation in patients with severe UC ranges from 21% to 34% [42, 43]. Since May 2018, the parameters for testing of CMV have been revised twice, continuously strengthening the specificity and accuracy of virus detection to reduce the possibility of re-exacerbation.” (Lines 133–136)

Question: English language should be significantly improved as some sentences may be confusing.

Response: We would like to thank the reviewer for evaluating our manuscript and for his/her comment. Please note that we have sent our manuscript to an English editing company (Editage) for English language proofreading. We hope that the language quality has significantly improved in the revised manuscript.

Question: Clostridium difficile should be replaced by Clostridioides difficile.

Response: We apologize to the reviewer for the mistake. Please note that we have corrected this mistake throughout the whole manuscript.

Question: Table 4 and 5 are very difficult to read and should be simplified.

Response: We would like to thank the reviewer for the suggestion. Please note that we have moved original Tables 4, 5 to supplemental material (Supplemental Tables 1, 2) and simplified them in the revised manuscript.

Question: "Other common enteric pathogens" screened in feces (Table 2) should be specified.

Response: We would like to thank the reviewer for the suggestion. Please note that we were referring to any pathogen found in fecal bacterial cultures that can cause intestinal disease, including vancomycin-resistant enterococcus, methicilin-resistant Staphylococcus aureus, arbapenem-resistant Enterobacteriaceae, and multidrug-resistant Gram-negative bacteria. We have included these pathogens in Table 2 of the revised manuscript.

Question: Many redundancies are present in the discussion and especially, the paragraph about the cost of donor screening may benefit from a rewriting.

Response: We would like to thank the reviewer for the insightful comment. Please note that we have revised the flow and content of the Discussion section, as per the reviewer’s suggestions. Especially, we have removed the paragraph stating the treatment cost. As our aim was to present our DS criteria and related data for reference, the intention was not to compare whether the increased cost will bring benefits or to present its advantages and disadvantages compared to other countries. Therefore, we decided to remove this paragraph to make our Discussion section more concise. We have rewritten the first paragraph to further clarify the purpose of our study, and revised the first paragraph as follows:

Since 2014, no serious adverse events have been noted in patients admitted to our studies, regardless of the use of 2014 or 2018 version of the DS criteria. However, other institutions have reported serious adverse events due to insufficient DS [16,17]. Therefore, it is necessary to continuously strengthen the screening of infectious diseases and minimize the occurrence of adverse events. To our best knowlege, this is the first report in Japan that describes the screening items, costs, and results to provide reference for future studies.(lines 213–219)

We have also adjusted the redundant and confusing parts of the limitations in the Discussion section. The revised part is as follows:

There were limitations to this study. First, we did not record the reasons why candidates were excluded at the preliminary assessment stage. Second, this was a single-center clinical study, and volunteers were recruited non-randomly. Third, the current parasite screening guidelines may cause unnecessary exclusion of candidates; thus, pass rates may not be representative. (lines 257–261)

Question: Authors mentioned "the expected efficacy" of FMT in UC, however exact FMT procedure and efficacy are still matter of debate and intensive research. This sentence may be nuanced.

Response: As per the reviewer’s suggestion, we have revised this part as follows:

“Since 2014, we have been studying the implementation of FMT in the treatment of UC and have reported the efficacy and safety. [24-26]” (Lines 62–63)

Reviewer 2 Report

This manuscript showed the application of new donor screening criteria for fecal microbiota transplant. The text is clear, logic and well written in the introduction, methods and results. However this manuscript is descriptive, the aim is not clear and not adequately discussed. It is not clear what brought the introduction of more stringent criteria, except  for the increase in the cost of the screening and the reduction of the number of eligible donors.

Other observation:

-line 50. You cited a study in which two cases of insufficient screening were reported. In the study one patient died due to bacteremia caused by drug-resistant E. coli.  Why don't you have included in the criteria that you propose in this manuscript a screening for gram negative MDR, for Enterococcus VR or for MRSA?  Since other DS criteria screen this colonization, the reason of your choice should be explained more extensively than at line 52.

Author Response

Question: However this manuscript is descriptive, the aim is not clear and not adequately discussed. However this manuscript is descriptive, the aim is not clear and not adequately discussed. It is not clear what brought the introduction of more stringent criteria, except for the increase in the cost of the screening and the reduction of the number of eligible donors.

Response: The authors would like to thank the reviewer for his/her constructive critique to improve the manuscript. We have made every effort to address the issues raised and to respond to all comments. The revisions are indicated in red font in the revised manuscript. Please, find next, a detailed, point-by-point response to the reviewer's comments. We hope that our revisions would meet the reviewer’s expectations.

In fact, none of the patients developed serious adverse events after using the 2014 version DS criteria or the 2018 version DS criteria in our hospital. However, other institutions have reported cases of development of serious adverse events due to inadequate DS criteria. Therefore, it is necessary for us to continuously strengthen the screening of infectious diseases to further prevent any possibility of developing serious adverse events.

Therefore, our aim was to present our nearly 8-year process of strengthening the DS criteria for providing reference and share our experience, such as screening items, costs, and results. We did not aim to demonstrate what advantages or changes the stricter DS criteria would bring.

Therefore, please note that we have revised the flow and content of the Discussion section for potentially confusing aim. For example, We have rewritten the first paragraph to further clarify the purpose of our study, and revised the first paragraph as follows:

Since 2014, no serious adverse events have been noted in patients admitted to our studies, regardless of the use of 2014 or 2018 version of the DS criteria. However, other institutions have reported serious adverse events due to insufficient DS [16,17]. Therefore, it is necessary to continuously strengthen the screening of infectious diseases and minimize the occurrence of adverse events. To our best knowlege, this is the first report in Japan that describes the screening items, costs, and results to provide reference for future studies.(lines 213–219)

And we also restructured the content of research limitations and removed redundant parts in the discussion section, especially the discussion of cost.

Besides, we have revised the title to match the purpose and content of our study. The revised title is as follows:

“Donor Screening Revisions of Fecal Microbiota Transplantation in Patients with Ulcerative Colitis” (lines 1–3)

Question: line 50. You cited a study in which two cases of insufficient screening were reported. In the study one patient died due to bacteremia caused by drug-resistant E. coli.  Why don't you have included in the criteria that you propose in this manuscript a screening for gram negative MDR, for Enterococcus VR or for MRSA?  Since other DS criteria screen this colonization, the reason of your choice should be explained more extensively than at line 52.

Response: We would like to thank the reviewer for the valuable comments. We would like to apologize for the confusion. In fact, since 2014 we have been checking for pathogens found in fecal bacterial cultures that can cause intestinal disease, including vancomycin-resistant enterococcus, methicilin-resistant Staphylococcus aureus, arbapenem-resistant Enterobacteriaceae, and multidrug-resistant Gram-negative bacteria. We collectively referred to them as “other common enteric pathogens.” We have also filled the corresponding content in Table 2.

And following the FDA safety alert, we started checking all E. coli strains since August 2020. Moreover, we included drug-resistant E. coli. The revised part is as follows:

“In addition, E. coli verotoxin testing has been performed since June 2019, and further screening of all E. coli strains has been performed promptly from August 2020 because of safety alerts (March 12, 2020) issued by The U.S. Food and Drug Administration (FDA) [37].” (Lines 114–117)

Round 2

Reviewer 1 Report

The authors adressed most of my suggestions. 

However as previously suggested I would remove in the abstrac the concept of "expected efficacy" as FMT in UC is still under on-going evaluation

"Since 2014, FMT has been utilized to treat ulcerative colitis (UC) in our hospital and has achieved the expected efficacy without serious adverse events"

Author Response

Question: However as previously suggested I would remove in the abstrac the concept of "expected efficacy" as FMT in UC is still under on-going evaluation

Response: We apologize to the reviewer for the oversight. Please note that we have corrected this mistake in the abstract section and modified lines 18-19 of the abstract.

“Since 2014, FMT has been utilized to treat ulcerative colitis (UC) in our clinical studies and has shown efficacy and safety.”(lines 15-16)

“The donor candidates were screened according to the DS criteria.”(lines 18-19)
